# Molecular Characterisation of Antimicrobial Resistance in *E. coli* Isolates from Piglets in the West Region of Romania

**DOI:** 10.3390/antibiotics12101544

**Published:** 2023-10-15

**Authors:** Bianca Cornelia Lungu, Ioan Hutu, Paul Andrew Barrow

**Affiliations:** 1Horia Cernescu Research Unit, Faculty of Veterinary Medicine, University of Life Sciences King Michael I, Calea Aradului 119, 300645 Timisoara, Romania; bianca.lungu@fmvt.ro; 2School of Veterinary Medicine, University of Surrey, Daphne Jackson Rd., Guildford, Surrey GU2 7AL, UK

**Keywords:** piglets, *E. coli*, resistance genes, penetrance

## Abstract

Antibiotics are widely used for prophylaxis and therapy, reducing morbidity and mortality produced by bacterial pathogensin pigs, including infections caused by *Escherichia coli*. The aim of this study was to characterise antibiotic resistance phenotypes and genotypes in *E. coli* isolates in pigs in West Romanian grower farms. Differential phenotypic susceptibility profiles and the contribution of resistance genes to phenotypic expression of susceptibility or resistance were evaluated. A total of 76 *E. coli* isolates were identified and confirmed by the MicroScan Walk Away System. The occurrence of four resistance genes, ampC, blaZ, blaTEM and tetK in strains resistant to 13 antibiotics was assessed. Of the *E. coli* isolates, 0% showed resistance to meropenem, 3.9% to tigecycline and 10.5% to piperacillin/tazobactam, whereas, in contrast, 100% were resistant to ampicillin and mezlocillin, 76.31% to piperacillin and 59.3% to tetracycline. The prevalence of resistance genes in resistant isolates detected by q-PCR analysis was 97.0% for ampC, 96% for blaZ, 32.9% for blaTEM and 58.8% for tetK. Penetrance (the proportion of individuals carrying a particular variant of a gene that also expresses an associated trait) was 50% for ampC (32% for amoxicillin/clavulanate, 62% for cefazolin, 32% for cefepime, 100% for cefotaxime, 56% for cefuroxime and 99% for ampicillin), 65% for blaZ (32% for amoxicillin/clavulanate and 99% for ampicillin), 51% for blaTEM (81% for piperacillin) and 44% for the tetK gene (83% for tetracycline). The result of phenotypic antibiotic resistance testing may indicate the presence of plasmid-borne resistance, with a diagnostic odds ratio of a positive phenotypic resistance for tetK being 4.52. As a management decision, the maximum penetrance admitted for using a specific antibiotic for *E. coli* infections in pigs is recommended to be less than 20%.

## 1. Introduction

The West Romanian grower pig sector is subject to zoonotic restrictions as a result of huge losses from African swine fever, with associated pressure to increase economic efficiency and produce acceptable production indices in the face of increased mortality. A major cause of economic damage is produced by bacterial pathogens, including the various pathotypes of Escherichia coli, which produce a wide range of diseases in pigs, including neonatal and post-weaning diarrhoea (PWD) [1]. To this, we may add antimicrobial resistance (AMR), which has reached unacceptable levels in many sectors of livestock production in Europe [2,3] including Romania [4] and which can result in diseases being intractable to chemotherapy, with huge losses estimated in the EU at more than EUR 1.5 billion annually [5]. Globally, one report estimates that by 2050, human deaths could reach 10 million p.a. (392,000 in Europe), with a reduction in global GDP between 2 and 3.5% [6].

One of the most common pathotypes of *E. coli*, Enterotoxigenic *E. coli* (ETEC), colonises the duodenum and jejunum of the porcine small intestine using adhesive pili and secreting cholera-like toxins which induce fluid secretion resulting in diarrhoea [7]. Strategies commonly used to prevent and control neonatal colibacillosis aim at reducing the number of pathogenic *E. coli* in the environment by implementing hygienic measures and biosecurity. Additional approaches include active and passive immunisation, dietary supplements involving probiotics and prebiotics and genetic selection for ETEC-resistant lines [8]. However, antibiotics continue to be extensively used, therapeutically, prophylactically and metaphylactically, administered by the parenteral and oral routes and in water [1,9]. The tonnage of antibiotics used in the livestock sector in Romania, at 59.0 mg/PCU, remains less than 84.4 mg/PCU, which is the average in the 31 EU countries [10]. In 2021, the EU/EEA population-weighted mean consumption of antibacterials for system use in the community was 15.0 DDD per 1000 inhabitants per day, ranging from 7.2 in Austria to 24.3 in Romania (Figure 1) [11].

Antibiotic misuse and underuse, but also their use at recommended levels, can all favour the selection of resistant bacteria [12], much of the resistance being encoded by self-transmissible plasmids.

In many countries outside Europe and North America, antimicrobials are also still used for the growth stimulation of pigs and poultry, with the result that >90% of strains of *E. coli* from Chinese farms can show resistance to common antibiotics such as tetracyclines, sulphonamides, penicillins and aminoglycosides [13,14,15]. AMR thus remains a global problem. Following international calls for reduction in antimicrobial use and improved diagnosis [16,17,18], levels of resistance have begun to fall, with some broiler production, at least, now being possible without recourse to antimicrobial administration. This is clearly also desirable for pig production, although greater attention to biosecurity will be needed for this goal to be attained. Levels of resistance in pig rearing in parts of Europe remain high, although welfare-label production systems generally have a significantly lower antibiotic usage than conventional indoor systems, with the greatest difference being between organic and conventional weaners, irrespective of country [19,20,21,22,23]. The situation in Romania compares well with other EU countries [4,24,25].

To begin to understand the extent of the AMR problem in West Romania in growing pigs, we characterised antibiotic resistance phenotypes and genotypes in *E. coli* isolates in West Romania’s growing pigs. We investigated genes normally associated with plasmid carriage in terms of differential phenotypic resistance profiles and evaluated the contribution of selected resistance genes to the phenotypic expression of susceptibility and resistance.

## 2. Results

### 2.1. Microbiological Antibiotic Resistance/Susceptibility Testing (AST)

Of the 140 presumptive *E. coli* isolates, 54.3% (*n* = 76) were confirmed as *E. coli*. A total of 18 were identified from samples taken in the first quarter, 20 in the second quarter, 17 in the third and 21 in the last quarter of the study period. There was no significant difference between the number of identifications made in the different seasons (chi-squared value = 1.151, *p* = 0.765) or different farms (chi-square valued = 2.072, *p* = 0.558).

For the 76 isolates, 13 of 30 antibiotics were considered relevant for the resistance genes tested (ampC, blaZ, blaTEM and tetK). The results of the outputs of the MicroScan Walk Away 40 SI are shown, presented as MIC values, in Table 1.

High proportions of *E. coli* strains were susceptible to meropenem (100%), tigercycline (96.05%) and piperacillin/tazobactam (89.47%). High prevalence values for resistance to ampicillin and mezlocillin (100%), piperacillin (76.31%) and tetracycline (67.1%) were recorded.

### 2.2. Prevalence of Resistance Genes

The prevalence of the resistance genes ampC, blaZ, blaTEM and tetK against the tested antibiotics, detected in susceptible and resistant isolates, are presented in Table 2. The prevalence of resistance genes (RG+ and SG+) in resistant (R) and susceptible (S) isolates, the penetrance of the genes (P%) and diagnostic odds ratios of positive phenotypic resistance (DOR) for *E. coli* in growing pigs are shown. This showed that 62.8% (468/745) of the 76 isolates tested for the 13 antibiotics and possessing the ampC, blaZ, blaTEM and tetK genes (RG+) actually showed the resistant phenotype (R) in AST, identified by MicroScan Walk Away. More importantly, 63.0% (392/622) of the isolates possessing the genes studied (SG+) actually showed the susceptible phenotype (S) in AST. The genes studied were found in 62.9% (860/1367) of R&S isolates. Only 37.2% (277/745) of isolates with the resistant phenotype (R) did not possess the genes studied (RG-) in AST, and 37.0% (230/622) of isolates with the susceptible phenotype (S) did not possess the genes studied (SG-). Some isolates showed the multi-resistance genes ampC and blaTEM (resistance to cefazolin, cefepime, cefotaxime, cefoxitin, ceftazidime, cefuroxime) and other isolates showed the multi-resistance genes ampC, blaZ and blaTEM (resistance to amoxicillin/clavulanate and ampicillin).

### 2.3. Microbiological Antibiotic Susceptibility Testing (AST) by Resistance Genes

The prevalence of resistance genes for phenotypic resistance and the susceptibility tothe antibiotics tested are presented in Table 3. The prevalence of ampC was 97.0%, the prevalence of blaZ was 96.0%, the prevalence of blaTEM was 32.9% and the prevalence of tetK was 58.8%.

### 2.4. The Association of Resistance Genes with Reduction in Susceptibility

The association between the possession of resistance genes (RG+) and reduced susceptibility (S) was analysed. Odds ratios showed that reducing susceptibility increased the risk of susceptible isolates carrying resistance genes. Gene carriage correlated significantly with reduced susceptibility in the tetK gene (OR = 2.85, *p* = 0.004, as shown in Table 4). The results of the study for the ampC, blaZ and blaTEM genes were not statistically significant.

### 2.5. Penetrance of Plasmid-Borne Genes

To estimate the penetrance of plasmid-borne genes, we eliminated the isolates with phenotypic resistance but not carrying plasmid-borne genes on the basis that the resistance would probably have been due to chromosomally encoded genes or genes encoded by other plasmid-borne genes that were not considered in this study. The estimated penetrance of the genes as per the interpretations of AST is shown in Table 2 as P% and summarised in Table 3.

For ampC, the overall penetrance was 50%, meaning that 50% of *E. coli* strains showing phenotypic resistance possessed the ampC gene (Table 2). For the antibiotics involved in resistance produced by the ampC gene, penetrance showed the following values: for amoxicillin/clavulanate, the penetrance was 32%; for cefazolin, 62%; for cefepime, 32%; for cefotaxime, 100%; for cefoxitin, 21%; for ceftazidime, 2%; for cefuroxime, 56%; and for ampicillin, 99%.

In the case of blaZ, the overall penetrance was 65%. For the antibiotics involved in resistance produced by the blaZ gene, penetrance showed the following values: for amoxicillin/clavulanate, 32% and for ampicillin, 99% (Table 2).

For blaTEM, the overall penetrance was 51%. For the antibiotics involved in resistance produced by the blaTEM gene, the greatest risk was seen for piperacillin, for which the penetrance value was 81% (Table 2).

For tetK, the overall penetrance was 44%. For the antibiotics involved in resistance produced by the tetK gene, the greatest risk was seen for tetracycline, for which the penetrance value was 83% (Table 2).

### 2.6. AST Diagnostic Odds Ratio of Positive Phenotypic Resistance (DOR)

To determine how the result of phenotypic AST could indicate the presence of plasmid-borne resistance, we calculated the diagnostic odds ratio of a positive phenotypic resistance, as shown in the DOR column in Table 4. A value greater than 1 shows that phenotypic resistance is effective in detecting isolates with genes inducing resistance to antibiotics such as tetracycline (OR = 4.52), piperacillin/tazobactam (OR = 3.73), tigecyline (OR = 2.29), piperacillin (OR = 1.48) and amoxicillin/clavulanate (OR = 1.24).

## 3. Discussion

This study was unable to identify any seasonal effects, either in percentages, prevalence of resistance or susceptibility, penetrance or the diagnostic odds ratio of a positive phenotypic resistance. This is quite normal because seasonal effects are minimised in intensive management systems and no modification was made during the study period to the standard operating procedures (SOPs) of treatments and prophylaxis of *E. coli* in the weaned piglets.

The results obtained here for the resistance genes ampC, blaZ, blaTEM and tetK in *E. coli* samples showed high levels of susceptibility to meropenem, tigecycline and piperacillin/tazobactam while, in contrast, all samples were resistant to ampicillin and mezlocillin, and more than 2/3 to piperacillin and to tetracycline. The prevalence of resistance genes in phenotypic resistant *E. coli* samples shown in q-PCR analysis was 97.0% for ampC, 96% for blaZ, 32.9% for blaTEM and 58.8% for tetK. As a result of the high levels of penetrance observed, we recommend that cefotaxime, ampicillin and tetracycline should no longer be used during the growth phase of pigs.

The use and overuse of antibiotics in water (and/or in feed) remains the main selective pressure for the development of phenotypic resistance. The overuse of ampicillin and tetracycline is common in many countries, either for therapy or, more commonly, prophylaxis, contrary to European Union rules on farm antibiotic use [26]. These two antibiotics are widely used globally to reduce morbidity and mortality caused by *E. coli* inpigs [27,28,29]. Zhong et al. (2022) [15] isolated *E. coli* from 1871 samples from pigs and their breeding environment in 31 Chinese provinces between 2018 and 2019, and found multi-resistance in 91% of *E. coli* isolates, including to ampicillin and tetracycline, but also to last-resort drugs including colistin, carbapenems and tigecycline. They also identified a heterogeneous group of O-serogroups and sequence types among the multidrug-resistant isolates which harboured multiple resistance genes, virulence factor-encoding genes and putative plasmids. In many countries outside Europe, there is no regulation of antibiotic use; colistin was regularly added to pig feed in China until the discovery of the plasmid-borne mcr-1 gene encoding colistin resistance, which prompted Chinese authorities to ban its use in animals [30]. By contrast, the European Union banned the prophylactic use of many antibiotics in food-producing animals in 2022 [31,32], as they did for growth-promoters in 2006.

Other authors [33] carried out a phenotypic and genotypic characterisation of antimicrobial resistance in German *E. coli* strains isolated from cattle, swine and poultry between 1999 and 2001. Resistance was found in 40% of the strains and multi-resistance was found in 32%. Resistance was significantly higher in isolates from poultry and swine than those from cattle. The most prevalent resistances were to sulfamethoxazole, tetracycline, streptomycin, ampicillin and spectinomycin. For each antibiotic, the predominant resistance genes were as follows: ampicillin, blaTEM1-like; chloramphenicol, catA and cmlA1-like; gentamicin, aac(3)-IV; kanamycin, aphA1; streptomycin, aadA1-like and strA/B; sulfamethoxazole, sul2, sul1 and sul3; tetracycline, tet(A) and tet(B); and trimethoprim, dfrA1-like, dfrA17 and dfrA12. Class 1 integrons were found in 30% of the strains. They carried dfrA1-aadA1a, aadA1a, sat1-aadA1a, dfrA17-aadA5, oxa1-aadA1a and dfrA12-aadA2. Eleven percent of the strains were resistant to nalidixic acid [34]. The prevalence of resistance genes in the swine sector in this study was 54.5% (48.2% [695/(748 + 695)]—last row in Table 3), somewhat lower than in the study by some authors [33] where the value was 60%, but the results for tetracycline or ampicillin were excessively large in our study.The resistance genes involved were also not comparable; however, the longitudinal study here gives baseline information on the magnitude of the resistance problem and its genetic background in contemporary Romanian *E. coliinthe* swine sector.

The effects of using antimicrobials in livestock have been studied by many authors over several decades [35,36,37]. In addition to the new EU rules banning the routine use of antibiotics and their use for prophylaxis, they should not be used for metaphylaxis unless there is a risk of spread of an infection or if the level of an infectious disease in the group of animals is high and where no other appropriate alternatives are available. The use or non-use of antibiotics in this way must thus be balanced against the risk of loss to the industry. Lugsomya et al. (2018) [34] tried to compare non-use with routine in-feed antimicrobial use in pigs and the effects on antimicrobial-resistant (AMR) *E. coli* at different stages of growth. A total of 300 commensal *E. coli* isolates were examined for AMR genes, plasmid replicons and molecular types; *E. coli* containing resistance genes were significantly increased in the nursery and growing periods in the farms where use was routine compared to the farms where there was no use.

Clearly, veterinary practitioners are frequently in a quandary over the dilemma of whether or not to use antibiotics. Use and overuse results in increased selection for antibiotic resistance but may increase animal survival, although the presence of resistance to the antibiotics used may reduce or even nullify efficacy as it must occur frequently in those countries where use is not regulated. For this issue, the penetrance and diagnostic odds ratio of a positive resistance phenotype may direct strategies for developing a programme of antibiotic use. At the farm level, when observing poor results of specific antibiotic treatments in infections with phenotypically sensitive *E. coli*, periodic screening for resistance genes is recommended in order to determine penetrance as an indicator of likely success.

Previous studies [38,39,40,41] have indicated the value of the multiple antibiotic resistance index (MARI) as a global measure of concern [32,35,36]. We found that this index of resistance correlated well (r = +0.979 at *p* = 0.000, unpublished results) with penetrance. In the case of high penetrance values, the relevant antibiotic should clearly not be used because of the risk from strains of *E. coli*, or other members of the Enterobacteriaceae, which are phenotypically susceptible but possess the resistance gene. In this situation (phenotypically susceptible but having the resistance gene, SG+), the current authors recommend, as a management decision, that the maximum penetrance permissible for using a specific antibiotic for *E. coli* infections in pigs should not be greater than 20%. However, future research will be needed to confirm this approach in treatment results using this level of penetrance and MARI values.

International institutions have recommended reduced use and increased regulation of use to stabilise the current global and regional situation regarding AMR. The use of penetrance as a predictive parameter guiding the selection of antibiotic for use where absolutely necessary will ensure improved targeting.

## 4. Materials and Methods

### 4.1. Description of Sampling Sites

The West region of Romania contains the largest number of swine farms in the lower plain areas. We studied four farms chosen from the Arad and Timis counties (Figure 1, upper left) chosen atrandom and under a confidentiality agreement. Samples were collected every three months (one sampling for the winter, spring, summer and autumn seasons) for a year. Young pigs showing signs of diarrhoea, between 7 to 30 kg in weight, were sampled by stratified sampling from three separate pens in each of the four farms.

### 4.2. Samples and E. coli Isolation

A total of 140 rectal faecal samples were collected from pens with diarrheic pigs [42]. About 10 g of faeces was collected with ESwab^TM^ (COPAN Diagnostics, Murrieta, CA, USA) and transported to the laboratory in a sterile pack. MacConkey agar (BIOLAB, Budapest, Hungary) was used for culture and presumptive identification with inoculation by swab and streaking out with incubation at 37 °C for 24 h.

Presumptive *E. coli* colonies were picked and purified on a MacConkey plate and identified by colonial morphology. A purified colony from each strain was transferred to 0.5 mL nutrient broth (BIOLAB, Budapest, Hungary) and stored at −80 °C for q-PCR analysis.

### 4.3. Confirmation of E. coli and Detection of Antimicrobial Resistance by MicroScan Analyser

Confirmation of identification of *E. coli* strains and detection of antimicrobial resistance or susceptibility by minimum inhibitory concentration (MIC) was carried out using the MicroScan Walk Away 40SI System (Dade Behring, West Sacramento, CA, USA), using the manufacturer’s instructions [43,44]. This automated process involves following the sample through the process with barcode allocation. Bacterial colonies were mixed with a diluent and transferred to MicroScan plates. The output classes were S (antibiotic-sensitive), I (intermediate) and R (antibiotic-resistant). Resistant strains were in a class formed by grouping together the intermediate and resistant classes.

### 4.4. DNA Extraction and Detection of Plasmid-Borne Resistance Genes by q-PCR

The DNeasy Blood & Tissue Kit (Qiagen, Hilden, Germany) was used for DNA extraction following the manufacturer’s instructions. DNA concentration was quantified using a Nano Quant Plate^TM^ (Tecan Trading AG, Männedorf, Switzerland), measuring absorbance at 260 nm/280 nm. For quantitative PCR reactions, 25 µL of the total reaction, 12.5 µL of SYBR™ Green Master Mix (Thermo Fisher Scientific, Waltham, MA, USA) and 1 µL of each forward (FW) and reverse (RV) primers for ampC, blaZ, blaTEM or tetK (Metabion International AG, Planegg, Germany) genes (Table 5) were mixed with 25 ng of bacterial DNA and water. The master mix consisting of SYBR Green, FW and RV primers was made for each primer set. The quantity of water added was adjusted according to the DNA concentration of the given sample.

The genes were selected relative to the antibiotics used most frequently in swine production and also to the existing antibiotics in the NBC 42 panel (Negative Breakpoint Combo Panel Type 42).

Agilent Technologies Stragene Mx3005P (Agilent Technologies Division, Model nr. 401513, Germany) was used for q-PCR reactions to detect plasmid-borne resistance genes. Any samples with amplification starting later than the 40th cycle of the annealing step of PCR reaction were considered negative. For practical purposes, the cut-off for expression of resistance was considered for samples with more than 12 cycles (Ct threshold cycles) and less than 40 cycles.

The q-PCR results of the isolates with a resistant phenotype (R and I class of MicroScan’s outputs) were divided into RG+ class, with evidence of resistance genes, or RG- class, in the absence of resistance genes. Strains that were phenotypically sensitive were classified as SG+ if the gene was present and SG- if the gene was absent.

### 4.5. Phenotypic and Genotypic Resistance

Because levels of resistance varied between strains generated by antibiotic resistance/susceptibility testing (AST), the prevalence of resistance or susceptibility (antibiotic resistance index (ARI)) was measured [48]. Penetrance was calculated in order to estimate the extent to which the bacteria carrying the resistance genes were phenotypically resistant. Mathematically, penetrance (P%) is the ratio of the number of individuals showing the resistance phenotype (R+I classes from MicroScan outputs) to the total individuals having the resistance genotype (RG+ class, after q-PCR analysis), expressed as a percentage [49,50], as shown below.


$$\mathrm{Penetrance}\;(P\%)=\frac{\mathrm{phenotypically}\mathrm{resistant}\mathrm{and}\mathrm{having}\mathrm{the}\mathrm{resistance}\mathrm{gene}\;(\mathrm{RG}+)}{\mathrm{phenotypically}\mathrm{resistant}\mathrm{and}\mathrm{having}\mathrm{the}\mathrm{resistance}\mathrm{gene}\;(\mathrm{RG}+)+\mathrm{phenotypically}\mathrm{susceptible}\mathrm{but}\mathrm{having}\mathrm{the}\mathrm{resistance}\mathrm{gene}\;(\mathrm{SG}+)}\times 100$$


To determine how phenotypic AST might indicate the presence of plasmid-borne resistance, we calculated the diagnostic odds ratio of a positive phenotypic resistance (DOR) [50], as shown below.


$$\mathrm{DOR}=\frac{\mathrm{phenotypically}\mathrm{resistant}\mathrm{and}\mathrm{having}\mathrm{the}\mathrm{resistance}\mathrm{gene}\;(\mathrm{RG}+)\times \mathrm{phenotypically}\mathrm{susceptible}\mathrm{and}\mathrm{not}\mathrm{having}\mathrm{the}\mathrm{resistance}\mathrm{gene}\;(\mathrm{SG}\text{-})}{\mathrm{phenotypically}\mathrm{resistant}\mathrm{but}\mathrm{not}\mathrm{having}\mathrm{the}\mathrm{resistance}\mathrm{gene}\;(\mathrm{RG}\text{-})\times \mathrm{phenotypically}\mathrm{susceptible}\mathrm{but}\mathrm{having}\mathrm{the}\mathrm{resistance}\mathrm{gene}\;(\mathrm{SG}+)}$$


### 4.6. Statistical Analysis

The statistical tests used were the Pearson’schi-squared test, paired *t*-test and Wilcoxon signed-rank test (non-parametric), performed using SPSS Statistics for Windows, Version 17.0. (Chicago: SPSS Inc., Chicago, IL, USA). A *p*-value of <0.05 was considered to be statistically significant.

## 5. Conclusions

A survey of antibiotic-resistant *E. coli* strains from pigs with diarrhoea in West Romanian farms has clearly shown high levels of resistance to some antibiotics. The correlation of phenotypic and genotypic resistance with a limited number of plasmid-mediated resistance genes points to the value of penetrance (percentage of strains showing resistance as a phenotype compared with the number showing the resistance genotype) as a parameter which may be used as guidance for a more accurate targeting of chemotherapy for *E. coli*.

## Figures and Tables

**Figure 1 antibiotics-12-01544-f001:**
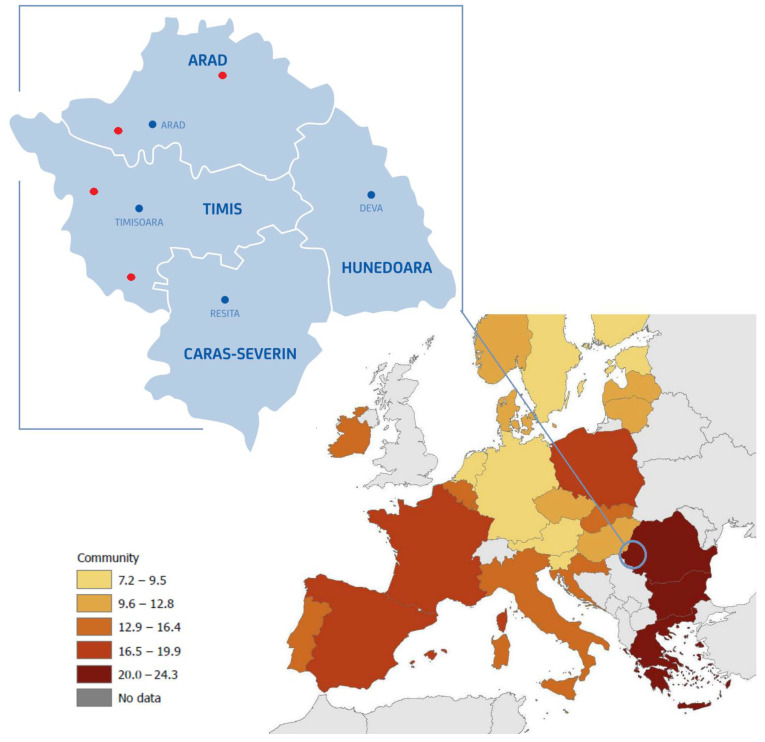
Community consumption of antibacterials for systemic use in EU/EEA countries in 2021 (right) [11] and the geographical area of the study (upperleft—red dots).

**Table 1 antibiotics-12-01544-t001:** The results of antibiotic susceptibility testing (AST) ^1^ for *E. coli* (*n* = 76, *p* = 0.000).

Antibiotic	MIC (mg/L) for *E. coli*		Resistant	Susceptible
	Resistant	Susceptible		
Amox/K Clav	>16/8	≤4	24 (31.57%)	52 (68.42%)
2. Ampicillin	>16	≤8	76 (100%)	0 (0%)
3. Cefazolin	>16	≤8	47 (61.84%)	29 (38.15%)
4. Cefepime	>16	≤8	24 (31.57%)	52 (68.42%)
5. Cefoxitin	>8	≤8	16 (21.05%)	60 (78.94%)
6. Ceftazidime	>16	≤1	3 (3.94%)	52 (68.42%)
7. Cefuroxime	>16	≤4	44 (57.89%)	32 (42.1%)
8. Meropenem	>8	≤1	0 (0%)	76 (100%)
9. Mezlocillin	>64	≤16	76 (100%)	0 (0%)
10. Pip/Tazo	>64	≤16	8 (10.52%)	68 (89.47%)
11. Piperacillin	>64	≤16	58 (76.31%)	18 (23.68%)
12. Tetracycline	>8	≤4	51 (67.1%)	25 (32.89%)
13. Tigecycline	>2	≤1	3 (3.94%)	73 (96.05%)

^1^ Results obtained by the MicroScan Walk Away for 76 identified *E. coli* samples.

**Table 2 antibiotics-12-01544-t002:** Prevalence of resistance genes (RG+ and SG+) in resistant (R) and susceptible (S) *E. coli* strains, penetrance of the genes (P%) and diagnostic odds ratios of a positive phenotypic resistance (DOR).

Genes	Antibiotics	R	RG+	RG-	S	SG+	SG-	P (%)	DOR
ampC	Amoxicillin/clavulanate	24	24	0	52	52	0	32%	
	Cefazolin	47	45	2	29	28	1	62%	0.80
	Cefepime	24	23	1	52	50	2	32%	0.92
	Cefotaxime	4	4	0	0	0	0	100%	
	Cefoxitin	16	16	0	60	60	0	21%	
	Ceftazidime	1	1	0	51	51	0	2%	
	Cefuroxime	44	40	4	32	31	1	56%	0.32
	Ampicillin	75	75	0	1	1	0	99%	
	TOTAL ampC	230	223	7	225	221	4	50%	0.58
blaZ	Amoxicillin/clavulanate	24	23	1	52	50	2	32%	0.92
	Ampicillin	75	72	3	1	1	0	99%	
	TOTAL blaZ	99	95	4	53	51	2	65%	0.93
blaTEM	Amoxicillin/clavulanate	24	9	15	52	17	35	35%	1.24
	Ampicillin	76	26	50					
	Cefazolin	47	15	32	29	11	18	58%	0.77
	Cefepime	24	6	18	52	20	32	23%	0.53
	Cefotaxime	4	2	2					
	Cefoxitin	16	5	11	60	21	39	19%	0.84
	Ceftazidime	3	1	2	52	20	32	5%	0.80
	Cefuroxime	44	12	32	32	14	18	46%	0.48
	Meropenem				76	26	50	0%	
	Mezlocillin	76	26	50					
	Pip/Tazo	8	5	3	68	21	47	19%	3.73
	Piperacillin	58	21	37	18	5	13	81%	1.48
	TOTAL blaTEM	365	120	245	319	114	205	51%	0.88
tetK	Tetracycline	51	30	21	25	6	19	83%	4.52
	Tigecycline	3	2	1	73	34	39	6%	2.29
	TOTAL tetK	54	32	22	98	40	58	44%	2.11
	TOTAL STUDY ^1^	748	470	278	695	426	269	52%	1.07

^1^ Total isolates was 76 *E. coli* samples tested for 13 antibiotics (Table 1) and for 4 genes (first columns of Table 2).

**Table 3 antibiotics-12-01544-t003:** Prevalence of resistance genes in resistant and susceptible *E. coli* strains ^7^.

Genes	Prevalence of Resistance Genes/Resistant Isolates ^5^	Prevalence of Resistance Genes/Susceptible Isolates ^6^
ampC ^1^	223/230 (97.0%)	221/225 (98.2%)
blaZ ^2^	95/99 (96.0%)	51/53 (96.2%)
blaTEM ^3^	120/365 (32.9%)	114/319 (35.7%)
tetK ^4^	32/54 (59.3%)	40/98 (40.8%)

^1^ For 8 antibiotics^,^ presented in Table 3; ^2^ For 2 antibiotics, presented in Table 3; ^3^ For 12 antibiotics^,^ presented in Table 3; ^4^ For 2 antibiotics^,^ presented in Table 3. ^5^ Value of chi test *p* = 0.000, ^6^ Value of chi test *p* = 0.000, ^7^ Value of chi test for percentages *p* = 0.368.

**Table 4 antibiotics-12-01544-t004:** Association between the possession of resistance genes and reduced susceptibility.

Factor	Outcome Genes	Risk Estimate			Pearson Chi-Square (Significant Correlation (≤0.05))
		Odds Ratios (Increased Risk > 1)	95% Confidence Interval		
			Lower	Upper	
Reduced susceptibility	ampC	0.731	0.332	1.605	0.285
	blaZ	0.931	0.294	2.945	0.634
	blaTEM	0.935	0.792	1.104	0.240
	tetK	2.850	1.281	6.342	0.004

**Table 5 antibiotics-12-01544-t005:** Primers used for the amplification of resistance genes.

Gene	Primer	Primer Sequence (5′-3′)	Annealing Temperature (°C)	Amplicon Size (bp)	Authors
ampC	ampC F	ATCAAAACTGGCAGCCG	65	510	[45]
	ampC R	GAGCCCGTTTTATGCACCCA			
blaZ	blaZ F	ACT TCA ACA CCT GCT GCT TTC	60	490	[46]
	blaZ R	TGA CCA CTT TTA TCA GCA ACC			
blaTEM	blaTEM F	GAGTATTCAACATTTCCGTGTC	42	850	[45]
	laTEM R	TAATCAGTGAGGCACCTATCTC			
tetK	tetK F	TCG ATA GGA ACA GCA GTA	55	169	[47]
	tetK R	CAG CAG ATC CTA CTC CTT			

## Data Availability

All of the data presented were obtained from subjects involved in this study. The name of companies and farms are under confidentiality agreement.

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
