# Peer review of "Molecular Characterisation of Antimicrobial Resistance in E. coli Isolates from Piglets in the West Region of Romania"

_antibiotics, 2023, doi:10.3390/antibiotics12101544_

Round 1

Reviewer 1 Report

Dear authors, greetings!

The manuscript "Molecular characterization of antimicrobial resistance in E. coli isolates from piglets in the West Region of Romania" is dedicated to investigate the penetrance and odds ratio of positive phenotypic resistance to antibiotics in E. coli (containing or not resistant genes) isolated from feces of diarrheic pigs from farms of West Romania.

Some adjustments/improvements are necessary prior to publishing the results in "Antibiotics".

The abstract needs to be adjusted to present data regarding penetrance calculated and the importance of this to evaluate resistance.

Regarding abbreviations, in International System the L that stands for liter is a capital one. So mL needs to be written with a capital L. Please adjust it.

In Introduction, it is interesting to mention the amount of money associated to the loss due to antibiotic resistance. This will highlight the importance of studies such as the one developed in this manuscript.

When it comes to Materials and Methods, subsection 2.1 could benefit from a map indicating the approximate location of the farms. In subsection 2.2 it is interesting to define the correct time (18 to 24 hours). The time of incubation varied from one sample to another? If yes, please explain why. In subsection 2.4 it is necessary to pay attention to the abbreviation: RT q-PCR is performed based on RNA extraction and not DNA. Please explain the RNA extraction performed. If the material was DNA, instead, the correct abbreviation would be only q-PCR (real-time).

Regarding Results, how was the bacteria confirmed as being E.coli? It is necessary to explain that in Methods' section and present the results of the analysis (was it confirmed using ribossomal RNA?). Table 2 lacks statistics, please fix that. In line 170 p.5, why was Tazobactam (89.47%) highlighted and not Meropenem (100%)? Regarding resistance genes, were genes of resistance to different drugs present in the same sample (multi-resistance)? It would be an interesting result to present.

 When it comes to conclusion is necessary to rewrite as it needs to be supported by data collected dring experiments. Only E. coli was analysed. 

Regarding English language, it is necessary to review some aspects:

1) repetition of words or use of very similar words near to each other (for example: cause and caused in line 88 p.1; tested in lines 161 and 162 p.4);

2)typos: diarrhoea (line 90 p.1 - I believe the intention was to write diarrhea); wets (line 384 p.9- I believe the intention was to write West)

3) lack of spaces between words: E.colistrains (Table 3 p.6); E.coli91% (line 317 p.7); includingcolistin(line 319 p.7); whichharbored (line 267 p.8)

4) grammar issues: increased mortality and reduced TO PRODUCE ACCEPTABLE PRODUCTION INDICES (line 87 p.1)

Author Response

Dear authors, greetings!

The manuscript "Molecular characterization of antimicrobial resistance in E. coli isolates from piglets in the West Region of Romania" is dedicated to investigate the penetrance and odds ratio of positive phenotypic resistance to antibiotics in E. coli (containing or not resistant genes) isolated from feces of diarrheic pigs from farms of West Romania.Some adjustments/improvements are necessary prior to publishing the results in "Antibiotics".

A: Thank you for your appreciation. All suggested adjustments have been made, in accordance with your suggestions.

The abstract needs to be adjusted to present data regarding penetrance calculated and the importance of this to evaluate resistance.

A: Penetrance was calculated in order to estimate the extent to which the bacteria carrying the resistance genes were phenotypically resistant. Wehave added more data regarding the penetrance for specific antibiotics.

Regarding abbreviations, in International System the L that stands for liter is a capital one. So mL needs to be written with a capital L. Please adjust it.

A: The abbreviation for ml has been changed with mL.

In Introduction, it is interesting to mention the amount of money associated to the loss due to antibiotic resistance. This will highlight the importance of studies such as the one developed in this manuscript.

A: We have included the global and regional economic damage caused by antibiotic resistance as far as information is available plus a reference to the potential global harm.

When it comes to Materials and Methods, subsection 2.1 could benefit from a map indicating the approximate location of the farms.

A: We have added, in a Figure, the counties and the red spots in the West Region of Romania where the farms were located.

In subsection 2.2 it is interesting to define the correct time (18 to 24 hours). The time of incubation varied from one sample to another? If yes, please explain why.

A: No, the time was established for 24 hours. We have modified the text accordingly.

In subsection 2.4 it is necessary to pay attention to the abbreviation: RT q-PCR is performed based on RNA extraction and not DNA. Please explain the RNA extraction performed. If the material was DNA, instead, the correct abbreviation would be only q-PCR (real-time).

A: This is an abbreviation error. Apologies.

Regarding Results, how was the bacteria confirmed as being E.coli? It is necessary to explain that in Methods' section and present the results of the analysis (was it confirmed using ribossomal RNA?).

A: E.colistrain identification was confirmed by theMicroScan Walk Away40SI System, Thesystem estimates, for each sample, the probability of identification (only outputs with the probability greater than 99.5% were considered as confirmed identification).

Table 2 lacks statistics, please fix that.

A: The statistics have been added.

In line 170 p.5, why was Tazobactam (89.47%) highlighted and not Meropenem (100%)?

A: We have made the correction.

Regarding resistance genes, were genes of resistance to different drugs present in the same sample (multi-resistance)? It would be an interesting result to present.

A: Yes, part of the samples showed several resistance genes. Some isolates showed the multi-resistance genes blaTEM (no tampC) and mecA(resistance to Cefazolin, Cefepime, Cefotaxime, Cefoxitin, Ceftazidime, Cefuroxime) and other isolates show the multi resistance genes blaTEM, blaZ and mecA(resistance to Amox/K clav and Ampicillin).

When it comes to conclusion is necessary to rewrite as it needs to be supported by data collected dring experiments. Only E. coli was analysed. 

A: We have removed the part mentioning other pathogens

Comments on the Quality of English Language. Regarding English language, it is necessary to review some aspects:

1) repetition of words or use of very similar words near to each other (for example: cause and caused in line 88 p.1; tested in lines 161 and 162 p.4);

A: We have made the corrections.

2)typos: diarrhoea (line 90 p.1 - I believe the intention was to write diarrhea); wets (line 384 p.9- I believe the intention was to write West)

A: We have made the corrections.

3) lack of spaces between words: E.coli trains (Table 3 p.6); E.coli91% (line 317 p.7); including colistin (line 319 p.7); which harbored (line 267 p.8)

A: We have also corrected these errors.

4) grammar issues: increased mortality and reduced TO PRODUCE ACCEPTABLE PRODUCTION INDICES (line 87 p.1)

A: We have corrected this mistake.

Reviewer 2 Report

The authors have pointed out firstly the isolates and identified E. coli from pigs farms were confirmed by the MicroScan Walk Away System and then the antibiotic sensitivity test of these bacteria were assessed. The prevalence of four resistance genes were also evaluated by RT-qPCR. In addition, the penetrance of individuals carrying a particular variant of a gene that also expresses analysis in resistant isolates were calculated.

However, the manuscript needs a minor revision for few sentences, and the scientific names of bacterial genera and species should be printed in italics. In addition, major revision to confirmed that can mecA found in E.coli by previous references

Minor revision

Abstract:

·       Lines 17-21: Revise the sentences

1. Introduction:

·       Lines 38-41: Revise the sentences   

·       Lines 54-56: re-write the paragraph

·       Lines 69-74: Sentences are too lengthy. Revise these sentences

4. Material and Methods

2.2 Samples and E. coli isolation

·       Line 87: A total of 140 rectal..… instead of 140

·       Line 114: add the primers supplier

·       Line 117: If you design the primers, please mentioned in text or cited each primers

4.2. Statistical analysis

·       Would you mention the name of software or package you used for analysis

2. Results

·       Lines 176-179: Sentences are too lengthy. Revise these sentences

·       Line 216: Table 3 why the P(%) of the total blaZ is 44% from 32% and 99%

3. Discussion

·       Lines 248-256: Revise the whole paragraph and please don’t repeat the result (meaning number and %)

·       Lines 262-24: Revise the sentences

·       Lines 274-293: Revise the whole paragraph and please don’t repeat the result (meaning number and %)

The major revision is to confirmed that mecA found in E. coli by previous references

Good Luck

In some extend Quality of English Language is ok

Author Response

The authors have pointed out firstly the isolates and identified E. coli from pigs farms were confirmed by the MicroScan Walk Away System and then the antibiotic sensitivity test of these bacteria were assessed. The prevalence of four resistance genes were also evaluated by RT-qPCR. In addition, the penetrance of individuals carrying a particular variant of a gene that also expresses analysis in resistant isolates were calculated.

However, the manuscript needs a minor revision for few sentences, and the scientific names of bacterial genera and species should be printed in italics.

In addition, major revision to confirmed that can mecA found in E.coli by previous references

A: Thank you for your comments which are appreciated. In the lab we are working on AMR in both S. aureus and E. coli. All suggested adjustments and amendmentshave been made in according with your suggestion.

Abstract:

  • Lines 17-21: Revise the sentences

As a native English speaker I am unclear about this sentence. It reads fine to me. If there are any serious issues with this could the reviewer please explain in greater detail. Many thanks.

  1. Introduction:
  • Lines 38-41: Revise the sentences   

We have deleted a part of the modified sentence and hope that this reads more clearly now.

  • Lines 54-56: re-write the paragraph

This has been rewritten and we hope that this now reads clearly.

  • Lines 69-74: Sentences are too lengthy. Revise these sentences

We have split this sentence

  1. Material and Methods

2.2 Samples and E. coli isolation

  • Line 87: A total of 140 rectal..… instead of 140

A: We have made the correction.

  • Line 114: add the primers supplier

A: We have added this information.

  • Line 117: If you design the primers, please mentioned in text or cited each primers

A: We did not design the primers. They were bought from a German supplier.

4.2. Statistical analysis

  • Would you mention the name of software or package you used for analysis.

A: We have added the name of software.

  1. Results
  • Lines 176-179: Sentences are too lengthy. Revise these sentences
  1. This sentence has also been split into two sentences

  • Line 216: Table 3 why the P(%) of the total blaZ is 44% from 32% and 99%

A: We apologize for this mistake which has been corrected in the table and accompanying text.

  1. Discussion
  • Lines 248-256: Revise the whole paragraph and please don’t repeat the result (meaning number and %).

A: We have revised the paragraph.

  • Lines 262-24: Revise the sentences

These sentences have been revised.

  • Lines 274-293: Revise the whole paragraph and please don’t repeat the result (meaning number and %)

 A: We have also revised this paragraph.

The major revision is to confirmed that mecA found in E. coli by previous references

Thank you for this observation. I am afraid that we have made a mistake. All the collected and calculated data were for the blaTEM gene, not mecA (thinking back, probably the mistake was with the table which was previously used for S. aureus). We made the necessary modifications

Reviewer 3 Report

The study is scientifically sound and well presented. Antibiotic resistance  presents a huge burden in treating farm animals and studies addressing the phenotypic and genotypic traits of the organism helps in controlling the situation in a  better way  eventually minimizing antibiotic abuse. Deciding a prophylactic measure based on phenotypic aspects along with antibiotic penetration index will lead to better results .

Author Response

Thank you for your very positive comment which is greatly appreciated

Reviewer 4 Report

The authors aimed to study antibiotic sensitivity / resistance in bacterial isolates from pigs in Romania.

An initial comment is that the authors present the work as a large countrywide study, but nevertheless it is of local interest (only a part of the country and only four farms), of targeted group of animals (only growing pigs) and of limited bacteria (only E. coli). So, please tone down the text to be more precise with the work performed.

Major issues

Why not sampling clinically healthy pigs as well? They are also E. coli-carriers.

How did you carry out presumptive identification of the bacteria? Not mentioned at all.

Many genes fully relevant with the antibiotic resistance have not been studied. Please carry out further evaluation before resubmitting, otherwise the work is incomplete.

The number of E. coli isolates studied is very small and does not allow to draw sensible conclusions.

Minor issues

The Introduction is verbose and out of context. I suggest to discuss the benefits of monitoring for antibiotic resistance, as being more appropriate.

Please include a map of the region with the spots of the locations of the farms.

Please include further details about the PCRs (e.g., product size).

Please revise after carrying out further work which needs to include a) the study of more E. coli isolates (to become at least 200 in total) and b) the study of additional antibiotic resistance-related genes.

The revised manuscript needs to be re-evaluated from scratch.

Author Response

An initial comment is that the authors present the work as a large countrywide study, but nevertheless it is of local interest (only a part of the country and only four farms), of targeted group of animals (only growing pigs) and of limited bacteria (only E. coli). So, please tone down the text to be more precise with the work performed.

A: The West Region of Romania has the largest number of swine reared in the plain areas – more than 98% of pigs are kept by one company in 50 farms (8000 to 50.000 animals/farm) split in two counties (Arad and Timis) with more than 1 million hogs sacrificedper year. We studied four farms per season, chosen atrandom, by the study protocol established in accordance with the Toma et all, 1996 (Epidemiologie Applique) methodology. The precondition such as the number of farms in the specified area and the high prevalence of E. coli (no swine farms were free of E. coli, including coli-carriers animals) gave us the opportunity to use a smaller number of farms and samples. The focus was not to identify infections caused byE. coli but to investigate AMR, penetrance etc. in. E. coli in the farms of the same plant/company, under the same management/routine (treatments). We have added a map showing the territory of study.

Major issues

Why not sampling clinically healthy pigs as well? They are also E. coli-carriers.

A: The aim of the study was to investigate E. coli strains from diarrheic piglets (Onn farm the pigletpens with diarrheic faeces remain the main reason for treatment with antimicrobials). Our intention was not an epidemiological study of healthy animals vs. animals with illness.

How did you carry out presumptive identification of the bacteria? Not mentioned at all.

A:  As now indicated in greater detail in the Methods section, Mac Conkey Agarwas used for culture with inoculation by swab and streaking out with incubation at 37ËšC for 24 h.  Presumptive E. coli colonies were picked and purified on a MacConkey plate, identified by colonial morphology. Confirmation of E. coli anddetection of antimicrobial resistance or susceptibility by minimum inhibitory concentration (MIC) was carried out using the MicroScan Walk Away40SI System (Dade Behring, West Sacramento, CA)using the Manufacturer’s instructions.

Many genes fully relevant with the antibiotic resistance have not been studied. Please carry out further evaluation before resubmitting, otherwise the work is incomplete.

A:The study focused on the genes related toantibiotics which are currently in use in the farms of the company. Also, some antibiotics are related to those used in human medicine.

The number of E. coli isolates studied is very small and does not allow to draw sensible conclusions.

A: Followingepidemiogical rules a good sample has to be “random” and with "enough cases". For the study, those criteria were respected. Thus, the results of “the sample” are sufficient to be reportablerelated to the population of piglets from West Romania.

Minor issues

The Introduction is verbose and out of context. I suggest to discuss the benefits of monitoring for antibiotic resistance, as being more appropriate.

A: We have reduced the length of the introduction very slightly but we did want to ensure that the aim, main data and the conclusion with suggested approach for further investigation was included which did require a fairly long introduction. We apologize for this and hope that it is acceptable. It does seem clearly presented.

Please include a map of the region with the spots of the locations of the farms.

A: It has not been possible to do this because of a confidentiality agreement with the farm company. We have indicated the counties involved.

Please include further details about the PCRs (e.g., product size).

 A: We added the bp of the primers in Table no 1.

Please revise after carrying out further work which needs to include a) the study of more E. coli isolates (to become at least 200 in total) and b) the study of additional antibiotic resistance-related genes.

  1. Following the epidemiological rules in a “random” study, the number of studied isolates is sufficient for representative analysis and the conclucions drawn.

The revised manuscript needs to be re-evaluated from scratch.

Round 2

Reviewer 1 Report

Dear authors,

greetings!

As requested the manuscript was improved.

Author Response

Thank you very much!

Reviewer 2 Report

Dear Authors

I wonder if the authors support by evidence for the presence of the mecA gene in E. coli. The evidence, they supported was for S. aureus and not E. coli such as the reference 31 is supported that mecA present in  for S. aureus

Strommenger B, Kettlitz C, Werner G, Witte W. Multiplex PCR assay for simultaneous detection of nine clinically relevant 450 antibiotic resistance genes in Staphylococcus aureus. J Clin Microbiol. 2003 Sep;41(9):4089-94. doi: 451 10.1128/JCM.41.9.4089-4094.2003. PMID: 12958230; PMCID: PMC193808.

Good Luck

The quality of English Language is to some extend ok

Author Response

As we answered in response to your first observation (in relation to the major revision -  mecAfound in E. coli by previous references), I am afraid that we made a mistake. All the collected and calculated data were for the blaTEM gene, not mecA (thinking back, probably the mistake was with the table which was previously used for S. aureus). We have made the necessary modifications using "find and replace" ampC with blaTEM instead of mecA with blaTEM.

Thank you again!

Reviewer 4 Report

The work is limited:
only a part of the country, 
only a few farms on the region,
only animals with clinical signs,
only some AMR-related genes.

The lack of healthy control animals in particular makes the work erratic and incomplete.
I cannot support publication, but this only a recommendation from my part, I return an opinion of major revision and I leave the matter to the academic editor to decide.

Author Response

Thank you for you opinion.